# Exploring HbA1c variation between Australian diabetes centres: The impact of centre-level and patient-level factors

**Matthew Quigley**[1], **Arul Earnest**[1], **Naomi Szwarcbard**[1], **Natalie Wischer**[1,2], **Sofianos Andrikopoulos**[1,3], **Sally Green**[1], **Sophia Zoungas**[1,4]*

**1** School Public Health and Preventive Medicine, Monash University, Melbourne, Victoria, Australia, **2** National Association of Diabetes Centres, Sydney, New South Wales Australia, **3** Australian Diabetes Society, Sydney, New South Wales, Australia, **4** Diabetes and Vascular Medicine Unit, Monash Health, Clayton, Melbourne, Victoria, Australia

* Sophia.Zoungas@monash.edu

**Data Availability Statement:** The datasets used and/or analysed during the current study are securely housed at the School of Public Health and Preventive Medicine, Monash University,

## Abstract

### Background

Increasing global diabetes incidence has profound implications for health systems and for people living with diabetes. Guidelines have established clinical targets but there may be variation in clinical outcomes including HbA1c, based on location and practice size. Investigating this variation may help identify factors amenable to systemic improvement interventions. The aims of this study were to identify centre-specific and patient-specific factors associated with variation in HbA1c levels and to determine how these associations contribute to variation in performance across diabetes centres.

### Methods

This cross-sectional study analysed data for 5,872 people with type 1 (n = 1,729) or type 2 (n = 4,143) diabetes mellitus collected through the Australian National Diabetes Audit (ANDA). A linear mixed-effects model examined centre-level and patient-level factors associated with variation in HbA1c levels.

### Results

Mean age was: 43±17 years (type 1), 64±13 (type 2); median disease duration: 18 years (10,29) (type 1), 12 years (6,20) (type 2); female: 52% (type 1), 45% (type 2). For people with type 1 diabetes, volume of patients was associated with increases in HbA1c (p = 0.019). For people with type 2 diabetes, type of centre was associated with reduction in HbA1c (p <0.001), but location and patient volume were not. Associated patient-level factors associated with increases in HbA1c included past hyperglycaemic emergencies (type 1 and type 2, p<0.001) and Aboriginal and Torres Strait Islander status (type 2, p<0.001). Being a non-smoker was associated with reductions in HbA1c (type 1 and type 2, p<0.001).

Melbourne, Australia. The data are not publicly available due to the inadvertent risk of identifying a participating diabetes centre (for example where there are limited numbers of participating sites within a given geographical area). Any requests for data should be directed to the ANDA Secretariat at anda@nadc.net.au. Any such requests will be forwarded to the ANDA Scientific Advisory Committee for consideration.

**Funding:** The authors received no specific funding for this work.

**Competing interests:** Sally Green is employed by Monash University and receives funding from the National Health and Medical Research Council (NHMRC), the Medical Research Future Fund (MRFF) and the Victorian Department of Health and Human Services. She has no declaration of interest specific to the research reported in this paper. Sophia Zoungas reports financial activities outside the submitted work including: Eli Lilly Australia Ltd – Participation in Steering Committee (CVOT) 2019 & 2021 on behalf of Monash University – payment to institution; Boehringer-Ingelheim – Participation in Advisory Board, expert committees or educational meeting 2019 on behalf of Monash University – payment to institution; MSD Australia - Participation in Advisory Board, expert committees or educational meeting 2019 & 2020 on behalf of Monash University – payment to institution; AstraZeneca – Participation in Advisory Board, expert committees or educational meeting 2017, 2018, 2019, 2020 & 2021 on behalf of Monash University – payment to institution; Novo Nordisk - Participation in Advisory Board, expert committees or educational meeting 2016, 2018, 2019 & 2020 on behalf of Monash University – payment to institution; Sanofi – Participation in Advisory Board, expert committees or educational meeting 2018 on behalf of Monash University – payment to institution. This does not alter her adherence to PLOS ONE policies on sharing data and materials. The other authors declare no relevant declarations of interest with regards to this manuscript.

**Abbreviations:** HbA1c, haemoglobin A1c; SBGM, self-blood glucose monitoring; BP, blood pressure; BMI, body mass index; ANDA, Australian National Diabetes Audit; NADC, National Association of Diabetes Centres; GDM, Gestational Diabetes Mellitus; CoE, Centre of Excellence; eGFR, estimated Glomerular Filtration Rate; CKD-Epi, Chronic Kidney Disease-Epidemiology; DKA, diabetic ketoacidosis; HHS, hyperosmolar hyperglycaemic state; CABG, coronary artery bypass grafting; LDL, low-density lipoprotein; HDL, high-density lipoprotein; SES, socioeconomic status; SEIFA, Socio-Economic Indexes for Areas.

## Conclusions

Centre-level and patient-level factors were associated with variation in HbA1c, but patient-level factors had greater impact. Interventions targeting patient-level factors conducted at a centre level including sick-day management, smoking cessation programs and culturally appropriate diabetes education for and Aboriginal and Torres Strait Islander peoples may be more important for improving glycaemic control than targeting factors related to the Centre itself.

## Introduction

Globally the rising incidence of diabetes is increasing the burden for patients and healthcare systems, in terms of both resource allocation and healthcare utilisation [1]. In addition, people with diabetes face added financial burden and the complexity of living with a chronic condition that often includes concomitant complications including psychological distress, retinopathy, neuropathy, nephropathy, amputation and increased risk of cardiovascular disease [2–4]. Optimal diabetes control requires people to self-manage multiple disease-influencing factors. These factors include diet, physical activity and long-term maintenance of blood glucose levels [5,6].

HbA1c (haemoglobin A1c), is the amount of glycated haemoglobin present in the blood, which increases with higher blood glucose levels. Measured by a blood test, HbA1C is commonly used as a measure of average glucose control over the few months prior to testing [7]. While self-blood glucose monitoring (SBGM) is used as a part of day to day self-management, HbA1c is the gold standard for evaluating overall diabetes control [8,9] with the literature recommending a HbA1c target of <7.0% (53 mmol/mol) for most people with diabetes [8,10–14].

In the UK and Australia, the following annual care practices are recommended for all people with diabetes: HbA1c; blood pressure (BP) monitoring; serum cholesterol; urine albumin/ creatinine ratio; foot risk surveillance; body mass index (BMI); smoking history; and digital retinal screening [15–19]. In Australia, whether or not these practices are being routinely delivered is measured by the Australian National Diabetes Audit (ANDA), an annual cross-sectional benchmarking activity of the National Association of Diabetes Centres (NADC), which documents the proportion of Australian people living with diabetes who a) receive these care practices and b) meet treatment targets.

Recent work from the UK suggests variation in practice and outcomes between diabetes centres, with the variation only partially explained by patient demographics [15]. However, treatment targets by locality appeared worse for people with type 1 diabetes and were not associated with patient demographics [15]. Similar differences by location have also been reported in Canada [20], the United States [21], and the Netherlands [22].

In Australia, ANDA has consistently shown mean HbA1c to be well above target for people with type 1 or type 2 diabetes [23–25]. Given the established links between elevated HbA1c and the risk of development of diabetes complications [4,26], lowering of HbA1c is an important marker of improvements in glucose control. Risk adjustment for patient-level factors outside the control of clinicians (such as age, duration of diabetes, or number of diabetes complications) has partially explained variation in outcomes such as HbA1c and blood pressure [27]. However, factors contributing to variation in HbA1c by type of diabetes centre or

location (metropolitan or regional) have not been explored. Identifying these factors may help inform the development and use of targeted interventions to help reduce such variation, with subsequent improvements in diabetes care and clinical outcomes.

## Aims and hypothesis

The aims of this study were to i) identify the centre-specific factors and patient-specific factors associated with variation in HbA1c levels, and ii) determine how these associations contribute to variation in performance for this clinical indicator across diabetes centres in Australia.

It was also hypothesised that different centre-specific factors and patient-specific factors are associated with variation in HbA1c for type 1 and type 2 diabetes.

## Methods

### ANDA administration and data collection

This was a cross-sectional study, with data collected during standard ANDA clinical audit. As per the ANDA protocol, the ANDA Secretariat invited diabetes centres in primary, secondary or tertiary care centres and specialist endocrinologists in private practice to participate in the ANDA collection. Participation was entirely voluntary and all contact and correspondence with participating centres/specialist endocrinologists occurred through the ANDA Secretariat. Other members of the ANDA team were blinded to the identity of individual sites, which were assigned a site code by the ANDA Secretariat [28].

During a four-week period in May-June 2019, de-identified data were collected for all consecutive patients presenting to one of 80 NADC-registered diabetes centres across Australia, using the standardised ANDA data collection form (S1 File). Use of this standardised form allowed collection of a minimum dataset that is congruent with similar international diabetes databases [28]. The data was collected during routine clinical consultations and involved review of the clinical record and pathology results where available. The data entry form was available to participating diabetes centres as a paper collection form, REDCap secure electronic data collection or secure data extraction from in-house databases. Where there were uncertainties regarding the data (such as extreme or illogical values), the ANDA data management team contacted the participating diabetes centre for clarification with erroneous data being removed from the dataset prior to analysis.

### Ethics approval and consent to participate

ANDA has received Human Research Ethics approval as an ongoing low risk clinical quality benchmarking activity, to use doubly de-identified data (participating site and individual patient) for research purposes (Monash Health Human Research Ethics Committee (HREC Reference number: HREC/17/MonH/123)). Verbal consent is obtained by the health practitioner at the time of clinical visit, where the purpose of the research is explained and participants are made aware that only deidentified information will be collected and used for research purposes. This research is carried out in accordance with the National Health and Medical Research Council (NHMRC) National Statement on Ethical Conduct in Human Research 2007 –updated 2018, and is pursuant with the low risk requirements therein [29].

### Participants

All people aged over 18 with type 1 or type 2 diabetes who presented to a participating diabetes centre and who had data for the dependent variable (HbA1c percentage) were included in this study. Participating ANDA collection centres primarily treat adults with type 1 or type 2

diabetes. As such, paediatric cases (i.e. <18 years of age) and people with Gestational Diabetes Mellitus (GDM) or unknown diabetes type were excluded.

There were 79 participating diabetes centres with eligible patients; these included a mix of Centres of Excellence (CoEs) (n = 5), Tertiary Diabetes Centres (n = 36), Secondary Care Diabetes Centres (n = 18) and Primary Care Diabetes Centres (n = 20), as defined by the NADC. All Centres of Excellence are Tertiary Diabetes Centres, but are also recognised for clinical, research, education, service advocacy and policy leadership. Centres of Excellence and Tertiary Diabetes Centres offer a suite of diabetes services with full time medical and allied health staff including endocrinologists, diabetes educators, psychologists, dieticians and podiatrists. Secondary Care Diabetes Centres employ a range of full/part time diabetes staff including a clinical lead, diabetes educators and dieticians, but may not employ an endocrinologist or other specialist staff. Primary Care Diabetes Centres employ diabetes educators and liaise with general practitioners. Due to the similarities between CoEs and Tertiary Diabetes Centres and the low numbers of CoEs, these centre categories were combined for analysis. Included cases and reasons for exclusion are shown in Fig 1. This research was reviewed by members of the ANDA Scientific Advisory Committee as per standard ANDA protocol [23].

## Variables

The dependent variable in the modelling was HbA1c percentage. Centre-level covariates were centre location (regional/metropolitan), centre type (Centres of Excellence and Tertiary Care Centres, or Secondary Care Centres, or Primary Care Centres) and number of patients. Continuous patient-level covariates were diabetes disease duration in years, the total number of glucose-lowering treatments and estimated Glomerular Filtration Rate (eGFR) (calculated according to the Chronic Kidney Disease Epidemiology (CKD-Epi) formula detailed by Levey et al. [30]). Categorical patient-level covariates were sex (male or female), Aboriginal or Torres Strait Islander status (yes/no), smoking status (yes/no), occurrence of severe hypoglycaemic episodes (yes/no), occurrence of recorded hyperglycaemic emergencies including diabetic ketoacidosis (DKA) and hyperosmolar hyperglycaemic state (HHS) (yes/no), occurrence of stroke or cardiovascular incidents including myocardial infarction, coronary artery bypass grafting (CABG)/angioplasty, or congestive cardiac failure (yes/no), the presence of any diabetes complications including retinopathy, peripheral neuropathy, ulceration, peripheral vascular disease, amputation, blindness, sexual dysfunction and end stage renal disease (yes/no), liver disease status (mild/moderate or severe/not applicable), age category (18–39 years, 40–59 years, 60–79 years and >80 years), and body mass index (calculated in kg/m$^2$ and categorised according to guidelines from the World Health Organization [31]: <18.5 (underweight); 18.5–24.9 (healthy); 25–29.9 (overweight) and >30 (obese)).

## Statistical analysis

A linear mixed effects model was used to identify factors associated with variation in HbA1c levels including the relative contribution of centre-level factors and patient-level factors. This was achieved by specifying a random intercept term for centre and fixed effects terms for all patient-level covariates. Univariate modelling was carried out to determine the most significant variables to include in the final models. Based on the most significant variable identified in the univariate analysis, we used the likelihood ratio test to evaluate whether the inclusion of the next most significant variable helped improve the fit of the model, and this was done sequentially until all potential variables were evaluated.

Separate models for type 1 and type 2 diabetes were developed as clinical rationale and previous studies suggested that different factors may contribute. Sensitivity analyses examined the

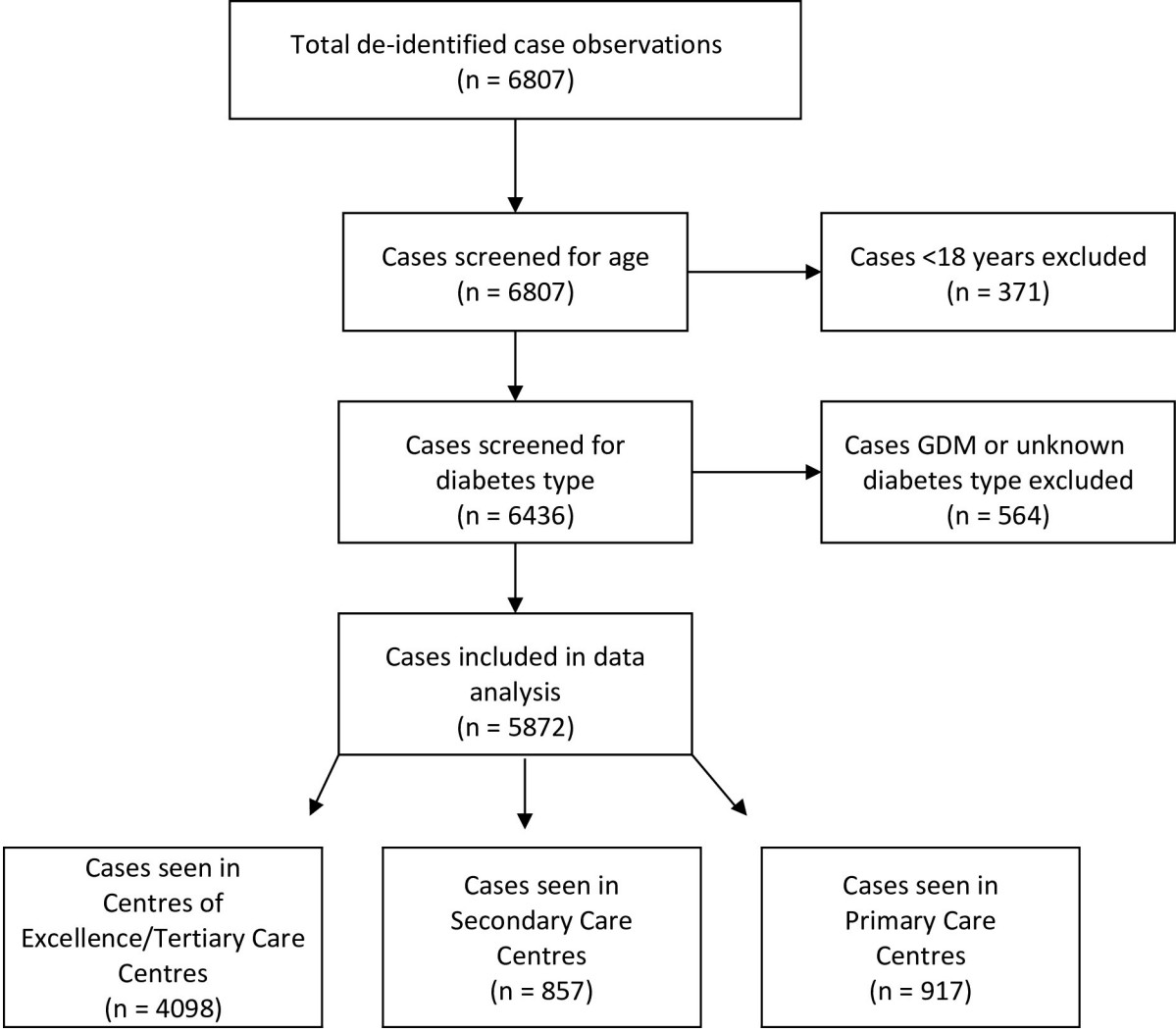

**Fig 1. Inclusion flow diagram.** Total de-identified case observations are shown, with screening and reasons for exclusion. Data included in the analyses are shown as per the treating diabetes centre type.

effect of excluding patients for whom data was collected at an initial visit to a diabetes centre, as clinical rationale would suggest that these patients may be newly diagnosed or referred to a diabetes centre due to difficulties in achieving optimal glycaemic control in which case HbA1c would be expected to be higher.

The level of significance was set at 5% and all data analysis was carried out in Stata, version 15 [32].

## Results

### Characteristics of the study population

Data from 5,872 patients with diabetes was analysed. People with type 1 diabetes and type 2 diabetes comprised 29% (n = 1,729) and 71% (n = 4,143) of the sample respectively, as per the clinical caseload of ANDA 2019 collection sites. The demographic and clinical data of the patients is shown in Tables 1 and 2.

**Table 1. Patient demographic and clinical variables–type 1 diabetes.**

| Demographic and clinical variables—type 1 Diabetes | | |
|---|---|---|
| N = 1,729 (% of total sample) | 29.44 | |
| Age in years (mean, SD) | | 43±17 |
| Disease duration in years (median, IQR) | | 18 (10,29) |
| Weight in kg (mean, SD) | | 80.65±18.83 |
| Height in metres (mean, SD) | | 1.70±0.11 |
| BMI[1] (mean, SD) | | 27.81±6.67 |
| Lipids -Total cholesterol in mmol/L (mean, SD) | | 4.65±1.08 |
| Lipids–LDL[2] cholesterol in mmol/L (mean, SD) | | 2.54±0.89 |
| Lipids–HDL[3] cholesterol in mmol/L (mean, SD) | | 1.51±0.46 |
| Lipids—triglycerides in mmol/L (mean, SD) | | 1.27±1.02 |
| Blood pressure—systolic in mmHg (mean, SD) | | 128.50±16.50 |
| HbA1c % (mean, SD) | | 8.4±1.7 |
| HbA1c mmol/mol (mean, SD) | | 68±19 |
| Sex | | |
| Female (%) | 51.89 | |
| Male (%) | 48.11 | |
| Smoking Status | | |
| Smoking–current (%) | 14.15 | |
| Smoking–past (%) | 22.19 | |
| Smoking–never (%) | 63.66 | |
| Aboriginal and Torres Strait Islander Peoples | | |
| No (%) | 97.43 | |
| Yes (%) | 2.57 | |

Table 1. Patient demographic and clinical variables–type 1 Diabetes. Characteristics of the sample with type 1 diabetes are shown.

[1]BMI: Body Mass Index (calculated in kg/m$^2$ and categorised according to guidelines from the World Health Organization [25]).

[2]LDL: Low-density lipoprotein.

[3]HDL: High-density lipoprotein.

## Factors contributing to variation in HbA1c

**Mixed effects modelling results—type 1 diabetes.** A higher volume of patients within each centre was associated with higher HbA1c levels among patients with type 1 diabetes. Centre location (regional/metropolitan) or centre type did not significantly contribute to variation in HbA1c (both p >0.05, Table 3).

Patient-level factors associated with higher HbA1c levels in type 1 diabetes included prior recorded hyperglycaemic emergency episodes, presence of diabetes complications and higher eGFR (all p < 0.01, Table 3). Patient-level factors associated with lower HbA1c levels in type 1 diabetes included non-smoking status and longer diabetes duration (all p < = 0.001, as shown in Table 3).

**Mixed effects modelling results—type 2 diabetes.** Being seen in a primary care centre was associated with lower HbA1c levels among people with type 2 diabetes (p = 0.001, Table 4). However, patient volume or centre location (regional/metropolitan) did not significantly contribute to variation in HbA1c (Table 4).

Patient-level factors associated with higher HbA1c levels in type 2 diabetes included the presence of diabetes complications (p = 0.016, Table 3), prior recorded hyperglycaemic

**Table 2. Patient demographic and clinical variables–type 2 diabetes.**

| Demographic and clinical variables—type 2 Diabetes | | |
|---|---|---|
| N = 4,143 (% of total sample) | 70.56 | |
| Age in years (mean, SD) | | 64±13 |
| Disease duration in years (median, IQR) | | 12 (6,20) |
| Weight in kg (mean, SD) | | 94.08±23.5 |
| Height in metres (mean, SD) | | 1.68±0.10 |
| BMI[1] (mean, SD) | | 33.34±7.80 |
| Lipids -Total cholesterol in mmol/L (mean, SD) | | 4.20±1.21 |
| Lipids–LDL[2] cholesterol in mmol/L (mean, SD) | | 2.16±0.94 |
| Lipids–HDL[3] cholesterol in mmol/L (mean, SD) | | 1.15±0.39 |
| Lipids—triglycerides in mmol/L (mean, SD) | | 2.24±2.21 |
| Blood pressure—systolic in mmHg (mean, SD) | | 133.01±17.50 |
| HbA1c % (mean, SD) | | 8.1±1.8 |
| HbA1c mmol/mol (mean, SD) | | 65±20 |
| Sex | | |
| Female (%) | 45.31 | |
| Male (%) | 54.69 | |
| Smoking Status | | |
| Smoking–current (%) | 11.49 | |
| Smoking–past (%) | 35.81 | |
| Smoking–never (%) | 52.71 | |
| Aboriginal and Torres Strait Islander Peoples | | |
| No (%) | 95.09 | |
| Yes (%) | 4.91 | |

Table 2. Patient demographic and clinical variables–type 2 Diabetes. Characteristics of the sample with type 2 diabetes are shown.

[1]BMI: Body Mass Index (calculated in kg/m$^2$ and categorised according to guidelines from the World Health Organization [25]).

[2]LDL: Low-density lipoprotein.

[3]HDL: High-density lipoprotein.

emergency episodes, Aboriginal and Torres Strait Islander status, and the number of glucose-lowering agents used (all p<0.001, Table 3). The only patient-level factor associated with lower HbA1c levels in type 2 diabetes was non-smoking status (i.e., being a non-smoker) (p <0.05, Table 4).

To identify the contribution of centre-level effects, the model for type 2 diabetes was examined both with and without the centre-level factors included. The addition of centre-level factors reduced the random intercept estimate from 0.265 to 0.185 (95% CI: 0.114 to 0.300) indicating that the variation in HbA1c not accounted for was substantively reduced by the addition of the centre-level factors.

**Sensitivity analyses.** ANDA collects a minimal cross-sectional dataset at one point each year. As such, the frequency of visits was not collected. However, whether the visit at which data was collected was an initial or subsequent visit was recorded. Clinical rationale suggested that HbA1c would likely be higher in people presenting to a health service for the first time, either because they had been recently diagnosed or referred to a specialist centre. In sensitivity analyses that excluded patients for whom data was collected at an initial visit to a diabetes centre, the same centre- and patient-level factors contributed to HbA1c variation. However, the

**Table 3. Mixed effects multivariate modelling of variation in HbA1c, type 1 diabetes.**

| Outcome variable: HbA1c percent | Coefficient | 95% CI | | P |
|---|---|---|---|---|
| **Centre-level factors** | | | | |
| Centre type (ref: COE[1] + Tertiary) | | | | |
| Secondary care | -0.071 | -0.347 | 0.205 | 0.616 |
| Primary care | 0.561 | -0.002 | 1.125 | 0.051 |
| Diabetes centre location (ref: metro) | 0.067 | -0.144 | 0.277 | 0.533 |
| Patient numbers (per 10 patient increase) | 0.010 | 0.000 | 0.002 | 0.019 |
| **Patient-level factors** | | | | |
| Diabetes duration (per 1-year increase) | -0.013 | -0.021 | -0.005 | 0.001 |
| eGFR[2] (per 1 mL/min/1.73m$^2$ increase) | 0.007 | 0.003 | 0.012 | 0.002 |
| Presence of diabetes complications (ref: no) | 0.502 | 0.279 | 0.725 | <0.001 |
| Smoking status (ref: current smoker) | -0.606 | -0.883 | -0.328 | <0.001 |
| Hyperglycaemic emergency episode (ref: no) | 0.623 | 0.420 | 0.827 | <0.001 |
| Age category (ref: 18–39 years) | | | | |
| 40–59 years | 0.162 | -0.081 | 0.405 | 0.191 |
| 60–79 years | 0.148 | -0.173 | 0.469 | 0.367 |
| > 80 years | 0.476 | -0.389 | 1.341 | 0.281 |
| BMI[3] category (ref: <18.49) | | | | |
| 18.5–24.99 | -0.471 | -1.177 | 0.235 | 0.191 |
| 25–29.99 | -0.661 | -1.368 | 0.046 | 0.067 |
| > 30 | -0.709 | -1.421 | 0.003 | 0.051 |

Table 3. Mixed effects multivariate modelling of variation in HbA1c, type 1 diabetes. The relative contribution of centre-level factors and patient-level factors to HbA1c variation is shown for people with type 1 diabetes.

[1]COE: Centres of Excellence.

[2]eGFR: estimated Glomerular Filtration Rate (eGFR) (calculated according to the CKD-Epi formula detailed by Levey et al. [30]).

[3] BMI: Body Mass Index (calculated in kg/m$^2$ and categorised according to guidelines from the World Health Organization [31]).

magnitude of the effect was slightly smaller for the centre-level factors (Table 5 in S2 File & Table 6 in S3 File)

## Discussion

This study aimed to identify the centre-level and patient-level factors associated with variation in HbA1c levels for patients with type 1 and type 2 diabetes. Only 2 centre-level factors i.e. patient volume and centre type were found to be significantly associated with variation in HbA1c levels. In contrast, many more patient-level factors were associated with variation in HbA1c including past hyperglycaemic events, smoking status, and Aboriginal and Torres Strait Islander status. These results suggest that greater improvements in HbA1c may be achieved by targeting patient-level factors rather than centre-level factors.

### Effect of centre-level factors on HbA1c variation

We were surprised that diabetes centre location was not associated with variation in HbA1c levels, in contrast to the findings from other countries [15,20–22]. This may, in part, reflect the higher concentration of ANDA collection sites in larger cities and a limited number of remote diabetes centres in the sample. The contribution of diabetes centre type to HbA1c variation in people with type 2 diabetes is an unexpected finding. This finding may likely be modified by another factor not controlled for in our model, such as socio-economic status. Another

**Table 4. Mixed effects multivariate modelling of HbA1c variation, type 2 diabetes.**

| Outcome variable: HbA1c percent | Coefficient | 95% CI | | P |
|---|---|---|---|---|
| **Centre-level factors** | | | | |
| Centre type (ref: COE[1] + Tertiary) | | | | |
| Secondary care | 0.046 | -0.306 | 0.398 | 0.798 |
| Primary care | -0.548 | -0.884 | -0.212 | 0.001 |
| Diabetes centre location (ref: metro) | -0.068 | -0.336 | 0.199 | 0.616 |
| Patient numbers (per 1 patient increase) | 0.001 | -0.001 | 0.003 | 0.383 |
| **Patient-level factors** | | | | |
| Diabetes duration (per 1-year increase) | 0.006 | -0.001 | 0.013 | 0.097 |
| Total glucose lowering treatments (per 1 treatment increase) | 0.331 | 0.267 | 0.395 | <0.001 |
| Hyperglycaemic emergency episode (ref: no) | 1.015 | 0.676 | 1.355 | <0.001 |
| Smoking status (ref: current smoker) | -0.516 | -0.704 | -0.328 | <0.001 |
| Aboriginal and Torres Strait Islander status (ref: no) | 0.542 | 0.240 | 0.845 | <0.001 |
| Presence of diabetes complications (ref: no) | 0.155 | 0.029 | 0.280 | 0.016 |
| Age category (ref: 18–39 years) | | | | |
| 40–59 years | 0.137 | -0.128 | 0.402 | 0.309 |
| 60–79 years | -0.236 | -0.503 | 0.031 | 0.083 |
| > 80 years | -0.168 | -0.506 | 0.170 | 0.329 |
| BMI[2] category (ref: <18.49) | | | | |
| 18.5–24.99 | -0.459 | -2.143 | 1.224 | 0.593 |
| 25–29.99 | -0.424 | -2.102 | 1.254 | 0.620 |
| > 30 | -0.396 | -2.071 | 1.280 | 0.643 |

Table 4. Mixed effects multivariate modelling of HbA1c variation, type 2 diabetes. The relative contribution of centre-level factors and patient-level factors to HbA1c variation is shown for people with type 2 diabetes.

[1]COE: Centres of Excellence.

[2] BMI: Body Mass Index (calculated in kg/m$^2$ and categorised according to guidelines from the World Health Organization [31]).

potential modifying factor may be that people with more stable type 2 diabetes are being managed in primary care, and that more difficult cases are being referred to a more specialised secondary or tertiary care centre. This is consistent with previously published work utilising ANDA data for people with type 2 diabetes, which found higher HbA1c for people with type 2 diabetes treated in tertiary care versus primary or secondary care [23]. The finding of patient volume associated with higher HbA1c levels in patients with type 1 diabetes should be interpreted with caution given that the coefficient represents a 0.1 increase in HbA1c percentage for every additional 100 patients. It is likely that centres with higher patient numbers are CoEs/Tertiary Care Centres, as these tend to be located in more densely populated areas and may see more clinically complex patients.

## Effect of patient-level factors on HbA1c variation

**Type 1 and type 2 diabetes.** *1. Hyperglycaemic emergencies and HbA1c variation.* The finding of an association between higher HbA1c levels and prior hyperglycaemic emergency (DKA/HHS) is consistent with that reported by other studies among adults with type 1 and 2 diabetes [33–36]. Diabetic ketoacidosis (DKA) and hyperosmolar hyperglycaemic state (HHS) are acute clinical hyperglycaemic emergencies. Although DKA is frequently seen in children and adolescents with type 1 diabetes at first presentation, it is also seen in adults. Common precipitating factors may include infection, missed medication, and acute medical events [37,38]. The finding of increased HbA1c variation in people who have documented episodes

of DKA or HHS is not surprising due to the hyperglycaemia associated with these states, but this finding emphasises the importance of regular blood glusose monitoring and education for those with higher HbA1c. Higher body mass index, low socioeconomic status or lower levels of health literacy may also lead to difficulties in preventing hyperglycaemia [38].

Given that the reported mortality of HHS is between 10 and 20%, effective prevention and treatment are essential, especially as diagnosis may be delayed due to the absence of ketoacidosis [39]. While our results reflect people who have sought emergency medical care to treat hyperglcaemia in the previous 12 months, it is likely that there are cases self-treated by individuals without emergeny intervention and some individuals experiencing multiple hyperglycaemic emergencies in any 12 month period. People living with diabetes may also not be aware of sick day mangement protocols to manage hyperglycaemic states before they become medical emergencies [40].

Our finding highlights the need for clinical systems to Identify patients who are at risk of, or who have experienced DKA or HHS, so that appropriate intensification of treatment and sick day management education can occur. Such education should be delivered at a centre-level for patients at risk, to enable effective self-medication during periods of hyperglycaemia and subsequently reduce the incidence of hyperglycaemic emergencies.

*2. The presence of diabetes complications and HbA1c variation.* The presence of diabetes complications was associated with higher HbA1c levels irrespective of diabetes type. As per the ANDA data collection form (S 1), diabetes complications were defined as the presence of any of the following: retinopathy, peripheral neuropathy, ulceration, peripheral vascular disease, amputation, blindness, sexual dysfunction or end stage renal disease, categorised as a binary variable (yes/no). The landmark Diabetes Control and Complications Trial (DCCT) and the long-term follow up Epidemiology of Diabetes Interventions and Complications (EDIC) study found reduced complications in people with type 1 diabetes who had intensive glycaemic control [4,26]. It is likely that the occurrence of diabetes complications is preceded by extended periods of suboptimal glycaemic control. Additionally, it is possible that as people with diabetes develop diabetes complications, clinical effort may be targeted towards minimising the day-to-day impact of the diabetes complications, rather than on achieving or maintaining ideal glycaemic control. Given the association between the presence of diabetes complications and HbA1c variation, it would be helpful to explore the relationship between HbA1c variation and individual complications with longitudinal data in a larger sample.

*3. Smoking status and Hba1c variation.* The association shown in our results between smoking status and variation in HbA1c levels for people with type 1 or type 2 diabetes is consistent with work that demonstrates poorer overall glycaemic control and higher HbA1c levels for people who smoke [41,42].

While smoking rates in the ANDA sample are similar to those in the general population, the harmful effects of smoking on diabetes complications cannot be overstated, especially when combined with hyperglycaemia [43,44]. Clinical management guidelines recommend that smoking status in people with diabetes should be assessed at every clinical visit, with advice and referral offered to those who do smoke [45,46]. Recent work has highlighted the safety and effectiveness of pharmacological and behavioural change techniques for smoking cessation in people who are motivated to quit [43,47].

Stopping smoking is often a gradual process, with multiple attempts and involves conscious decisions to change behaviour [48]. This process can be aided by smoking cessation programs that use a mixture of pharmacological therapies and behaviour change techniques [43,49]. For smoking cessation, successful interventions use behaviour change techniques including goal setting, tobacco use assessment, action planning and restructuring of the environment [50]. It is possible that the successful adoption of behaviour change strategies by people with diabetes

to facilitate stopping smoking may also transfer to other lifestyle factors that affect HbA1c, such as nutrition and physical activity.

The fact that similar reductions in HbA1C variation were seen for non-smokers in the ANDA sample regardless of diabetes type further supports the strong push towards promotion of smoking cessation programs as both a prevention and management strategy for people with diabetes, especially given the cost-effectiveness of such programs in people with diabetes [51].

## Type 1 diabetes

**1. eGFR and HbA1c variation.** Although statistically significant, the finding of eGFR being associated with small increases in HbA1c for people with type 1 diabetes should be interpreted with caution, given the low magnitude of the coefficient. Clinically, we would suggest that increased blood glucose monitoring may be beneficial for these patients to help manage diurnal blood glucose variability.

## Type 2 diabetes

**1. Aboriginal and Torres Strait Islander status and HbA1c variation.** Our finding of increases in HbA1c in Aboriginal and Torres Strait Islander peoples with type 2 diabetes is consistent with other work describing poorer diabetes outcomes in this population. While Aboriginal and Torres Strait Islander peoples comprise approximately 3% of the population of Australia, there are well documented disparities in health access and health outcomes compared to non-Indigenous Australians [52,53]. There is a higher prevalence of type 2 diabetes, and the associated complications are major contributors to increased mortality and a lower life expectancy among Aboriginal and Torres Strait Islander peoples [52,54]. While some Aboriginal and Torres Strait Islander peoples live in remote areas with less stable access to health care and medication, higher HbA1c levels may reflect a lack of engagement due to centres not providing culturally appropriate health care [53,55]. Given that approximately half of Aboriginal and Torres Strait Islander peoples live in urban communities, consideration should be given to the implementation of codeveloped resources and culturally and linguistically appropriate diabetes education programs that have shown community acceptance, rather than relying on standard methods of delivery. [54,56–60].

Delivery of culturally and linguistically appropriate diabetes education programs will also involve education of health care professionals to ensure that the diabetes care and education offered can be tailored to the specific cultural needs of the recipients. Design of programs to educate healthcare professionals at a diabetes centre level should also involve collaboration with Aboriginal and Torres Strait Islander healthcare workers to ensure that the training is relevant to the communities that healthcare professionals service. Due to the disparities in diabetes outcomes for this group of people living with diabetes, future work should examine the impact of culturally appropriate education for both healthcare providers and the people that they treat.

**2. Glucose-lowering medications and HbA1c variation.** The association between increasing numbers of glucose-lowering medications and increases in HbA1c variation for people with type 2 diabetes may seem counterintuitive. However, this is reflective of clinical practice where despite intensification of treatment and multiple pharmacological agents, some patients do not see clinically significant declines in HbA1c [61]. It may be that patients who require multiple glucose-lowering agents have progressive disease and hence hyperglycaemia is more difficult to manage [7,62]. This is consistent with current treatment algorithms that are reactive and suggests a more proactive approach to the management of type 2 diabetes,

with intensive multifactorial interventions including lifestyle and pharmacological changes required to see reductions in HbA1c [63].

Previous work has also shown that clinical inertia leads to the delay of treatment intensification in people with type 2 diabetes and contributes to delays in starting treatment with insulin [64]. While insulin is included as a glucose-lowering treatment in the ANDA data collection form, details of treatment intensification or the date of initiation are not collected, due to the nature of the minimal dataset. Clinical inertia may be playing a part in this sample, with some patients being prescribed further oral antihyperglycaemic medications, leading to delayed initiation with insulin. As such, multifaceted education to healthcare providers would likely be necessary to ameliorate this clinical inertia [64,65].

## Strengths and limitations

Strengths of this study include the number of patient level factors collected via a standardised data collection form as part of routine clinical audit practice and the participation of a broad cross-section of primary, secondary and tertiary care Australian diabetes centres, which are reflective of clinical practice. As such, this sample is likely to be representative of the clinical population seen in Australia.

A limitation is the inability to undertake further analysis related to centre-specific factors as we were restricted to the data routinely collected in the ANDA. Such centre-specific factors might include an adjustment for the number and type of specialist staff, funding model, or the impact of socioeconomic status (SES). While we could adjust for socio-economic disadvantage as per the Australian Bureau of Statistics Socio-Economic Indexes for Areas (SEIFA), this would be by the location of the diabetes centre, not location of individual patients. Given the wide geographical catchment area of many diabetes centres in Australia and the considerable variability in individual patient SES within these catchment areas, it is unlikely that such analysis would substantively add to our understanding.

A further limitation is that the diabetes centres were classified by their category within the NADC category model. These NADC categories of centre type represent the wide variety of diabetes centres across Australia, where people living with diabetes are treated in a range of primary, secondary and tertiary care settings. While providing an overview of the structure and the range of staff typically employed at each centre, discrete data about the numbers of staff, staff to patient ratio and model of care at each site is not collected as part of ANDA. Future research may be helpful in elucidating the association of these factors with clinical outcomes. Finally, due to the cross-sectional nature of the data, it was not possible to infer causality.

While the sample includes a higher percentage of people with type 1 diabetes than in the general community, it is important to remember that the sample reflects the clinical caseload of diabetes centres across Australia collected during routine clinical audit in a defined timeframe. While people with type 1 diabetes are often seen in diabetes centres for continuing care, it is likely that people with type 2 diabetes who have more stable control may be under the care of general practitioners and not specialised diabetes centres [23].

## Conclusion

Our results suggest that programs run at a centre-level but targeting patient-level factors rather than centre factors themselves may be more beneficial in reducing variation in HbA1c. These programs might include greater education about managing sick days to prevent hyperglycaemic emergencies, smoking cessation programs and the use of culturally and linguistically appropriate diabetes education programs for Aboriginal and Torres Strait Islander peoples

with diabetes. Given the mandated changes in healthcare delivery in many countries as a result of COVID-19, developers of such programs should consider making these programs available via virtual mediums to encourage uptake. Further research to identify other centre-level factors may aid in the development of future models to optimise clinical outcomes in diabetes centres. In particular, qualitative research may be helpful to understand the experience and context of contemporary diabetes care in a range of diabetes centres in both metropolitan and regional settings.

## Supporting information

**S1 File. ANDA data collection form 2019.** Supplied by Australian National Diabetes Audit (ANDA), Monash University, Melbourne, Australia.
(DOCX)

**S2 File. Sensitivity analysis, Table 5.** Sensitivity analysis, type 1 diabetes, excluding patients for whom the recorded visit was an initial visit. The relative contribution of centre-level factors and patient-level factors to HbA1c variation is shown for people with type 1 diabetes. [1]COE: Centres of Excellence. [2]eGFR: estimated Glomerular Filtration Rate (eGFR) (calculated according to the CKD-Epi formula detailed by Levey et al. [30]). [3] BMI: Body Mass Index (calculated in kg/m$^2$ and categorised according to guidelines from the World Health Organization [31]).
(DOCX)

**S3 File. Sensitivity analysis, Table 6.** Sensitivity analysis, type 2 diabetes, excluding patients for whom the recorded visit was an initial visit. The relative contribution of centre-level factors and patient-level factors to HbA1c variation is shown for people with type 2 diabetes. [1]COE: Centres of Excellence. [2] BMI: Body Mass Index (calculated in kg/m$^2$ and categorised according to guidelines from the World Health Organization [31]).
(DOCX)

## Author Contributions

**Conceptualization:** Matthew Quigley, Arul Earnest, Sally Green, Sophia Zoungas.

**Data curation:** Matthew Quigley.

**Formal analysis:** Matthew Quigley, Arul Earnest.

**Investigation:** Matthew Quigley, Naomi Szwarcbard, Natalie Wischer, Sofianos Andrikopoulos.

**Methodology:** Arul Earnest, Sally Green, Sophia Zoungas.

**Project administration:** Natalie Wischer, Sofianos Andrikopoulos.

**Supervision:** Arul Earnest, Sally Green, Sophia Zoungas.

**Visualization:** Matthew Quigley.

**Writing – original draft:** Matthew Quigley.

**Writing – review & editing:** Matthew Quigley, Arul Earnest, Naomi Szwarcbard, Natalie Wischer, Sofianos Andrikopoulos, Sally Green, Sophia Zoungas.

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
