## [Decision Letter · Decision Letter 0]

28 Oct 2021

PONE-D-21-14548Exploring HbA1c variation between Australian diabetes centres: the impact of centre-level and patient-level factorsPLOS ONE

Dear Dr. Zoungas,

Thank you for submitting your manuscript to PLOS ONE. Firstly, let me apologize for the extended delay in finding suitable reviewers able to contribute to the peer-review process. We are now able to provide the outcome of the review of your submission. While the reviewers expressed enthusiasm for your submission, it does not fully meet PLOS ONE’s publication criteria in its current form. Therefore, we wish to invite you to consider the comments of the reviewers and to submit a revised version of the manuscript that addresses the points raised.

Specifically, please pay particular attention to the comments raised regarding; greater justification for the approach to utilize HbA1C only; expand the descriptions for some of the variables indicated (including any new interpretations from this additional detail) and other points raised to expand the scope and insightfulness of the situational health care context.

We look forward to receiving your revised manuscript.

Kind regards,

Spencer D. Proctor, PhD

Academic Editor

PLOS ONE

Journal Requirements:

"SG is employed by Monash University and receives funding from NHMRC, MRFF, and the Victorian Department of Health and Human Services. She has no declaration of interest specific to the research reported in this paper. SZ reports payment to institution (Monash University) from Eli Lilly Australia Ltd, Boehringer-Ingelheim, MSD Australia, AstraZeneca, Novo Nordisk, Sanofi, Servier, for work outside the submitted work. The other authors declare no relevant declarations of interest with regards to this manuscript." 

We note that you received funding from a commercial source: Eli Lilly Australia Ltd, Boehringer-Ingelheim, MSD Australia, AstraZeneca, Novo Nordisk, Sanofi and Servier.

Reviewers' comments:

Reviewer's Responses to Questions

**Comments to the Author**

1. Is the manuscript technically sound, and do the data support the conclusions?

Reviewer #1: Partly

Reviewer #2: Partly

2. Has the statistical analysis been performed appropriately and rigorously? 

Reviewer #1: Yes

Reviewer #2: Yes

3. Have the authors made all data underlying the findings in their manuscript fully available?

Reviewer #1: Yes

Reviewer #2: Yes

4. Is the manuscript presented in an intelligible fashion and written in standard English?

Reviewer #1: Yes

Reviewer #2: Yes

5. Review Comments to the Author

Reviewer #1: n this cross sectional study authors analyzed a data of 1729 patients with type 1 diabetes and 4143 patients with type 2 diabetes collected through Australian National Diabetes Audit. The aim of this study was to identify the centre specific and patient specific factors associated with variation of hemoglobin A1C levels. Authors suggested interventions targeting patient level factors since patient level factors had higher impact on variability. Their finding was that centre location (Regional/Metropolitan) or centre type did not significantly contribute to the variation of hemoglobin A1C. However, in patients with type 1 diabetes, who are treated in centres with high volume patients had higher hemoglobin A1C. Patient level factors associated with higher HbA1C in type 1 diabetes included prior recorded hyperglycaemic emergency episodes, presence of diabetes complications, and higher GFR. Patient level factors associated with lower hemoglobin A1C in type one diabetes included none smoking status, longer diabetes duration.

In patients with type 2 diabetes, being seen in primary care centre was associated with lower HbA1C. Patient volume or centre location did not significantly contribute to HbA1C. Factors associated with higher HbA1C in patients with type 2 diabetes included presence of diabetes complication, prior recorded hyperglycaemic emergency, aboriginal and Torres Strait Islander status and number of glucose lowering agents. The only patient level factor associated with lower hemoglobin A1C in type 2 diabetes was non-smoking status.

Author brought up the important concept to identify patient specific factors to target delivery of care. However, clarification is needed for their data and interpretation of their findings.

1) Effect of centre-level factors on HbA1c variation: Authors have identified that one reason might be that patient with stable type 2 diabetes are managed in primary care and more difficult cases are being referred to more specialized secondary or tertiary care centre. Authors have also cautioned about their finding in people with type 1 diabetes that patient volume association with her hemoglobin A1C was because tertiary care centres are located in more densely populated areas, where they see clinically complex patients.

There is no clear description of what is the available service at present to support authors’ interpretation. There is no analysis or data on what kind of care is delivered in these centres according to Chronic Care Model such as self-management support, clinical information system, or decision support. Another important information that will strengthen the paper is the frequency of contact with the health care providers, as well as ratio of patients and providers rather than volume of patients only. This information might also give the answer why diabetes centre location was not associated with variation in HbA1c and authors finding is different than other study. (Page 16-Line249)

2)

Effect of patient-level factors on HbA1c variation

a) Hyperglycaemic emergencies and HbA1c variation: This paragraph is like a textbook. This needs revision to make relevant and explain authors finding.

b) The presence of diabetes complications and HbA1c variation: “The presence of diabetes complications was associated with higher HbA1c levels irrespective of diabetes type.” The logical explanation is higher HbA1C is associated with diabetes complication. It is important to identify what is the cause and what is the effect to appropriately plan care delivery.

c) Smoking status and HbA1c variation: Authors found that being a non-smoker was associated with reductions in HbA1c. People without diabetes in ANDA cohort also showed similar result. Authors finding sure emphasize importance of smoking cessation for everyone.

d) Aboriginal and Torres Strait Islander status and HbA1c variation; Since there is higher prevalence of type 2 diabetes in Aboriginal and Torres Strait Islander status, authors very appropriately suggested implementation of codeveloped resources and culturally and linguistically appropriate diabetes education programs. It will be important to know what is the current state of therapeutic relationship between the centers and these groups of patients. Is there any purposeful process of learning offered to the health-care workers to integrate specific contexts who are involved in care of these patients?

e) Glucose-lowering medications and HbA1c variation: It is not clear if insulin is included in the medications. Authors interpretation is “this is reflective of clinical practice where despite intensification of treatment and multiple pharmacological agents, some patients do not see clinically significant declines in HbA1c” (Page 20-line 330). Really? Could it be due to well-known providers’ “therapeutic inertia” - failure to initiate or intensify therapy in a timely manner resulting uncontrolled hyperglycaemia in patients with type 2 diabetes. At times multiple agents are added just to defer insulin initiation.

Reviewer #2: Exploring HbA1c variation between Australian diabetes centres: the impact of centre-level

and patient-level factors.

Summary:

The aim of the study was to identify centre- and patient- specific factors associated with variation in HbA1c, and determine if/how these associations contribute to

variation in improvement across diabetes centres in Australia. The authors

utilised data from the 2019 Australian National Diabetes Audit, a large cross-sectional study that with 5,872 persons with diabetes from 79 diabetes centres across Australia.

The main findings described by the authors include that only 2 centre-level factors (i.e. patient volume and centre type) were significantly associated with variations in HbA1c levels. In addition, the authors found that patient-level factors were also associated with variation in HbA1c including the number of past hyperglycaemic events, presence of Diabetes complications and Aboriginal and Torres Strait Islander status.

General:

The manuscript is well written and easy to follow. The data and findings are contextually appropriate for the journal. The choice of statistical modelling appears to be appropriate. There are some points raised below that could be addressed in an effort to provide further clarification and strengthen the rationale for the interpretation.

1. The authors have chosen to focus on HbA1C as the sole metric for quality of care and health improvement of those with diabetes. It was not (entirely) clear why the authors chose only HbA1C and this should better reconciled for the readership. Preferably please include (some pilot level) comparisons with fasting glucose, and/or post-prandial/diurnal glucose concentrations to demonstrate that HbA1C is the most sensitive marker in this cohort for the objectives. Are there validation data from this cohort elsewhere that the authors could utilize to strengthen this approach?

2. For the international readership, please provide some scientific basis for why Aboriginal and Torres Strait Islander Peoples would be relevant to identify in the cohort.

3. For clarification, please provide a more accurate description of the ‘Presence of diabetes complications’ as well as the extent of the ‘Hyperglycemic emergency episode(s)’, and include statements in the discussion and/or expand the interpretation based on this added information.

4. It is known that increased adoption of behavioral change techniques by individuals increase the likelihood of smoking cessation. It is also plausible that individuals that are motivated to know/learn about health improvement will also be more motivated to implement self -care strategies to improve diabetes status and manage both glucose and HbA1C. Please consider these points in the discussion.

5. Please consider moving the ethical statement earlier(up) in the methods section.

6. PLOS authors have the option to publish the peer review history of their article (what does this mean?). If published, this will include your full peer review and any attached files.

Reviewer #1: No

Reviewer #2: No

---

## [Author Response · Author response to Decision Letter 0]

19 Dec 2021

Response to reviewers.

Manuscript: Exploring HbA1c variation between Australian diabetes centres: the impact of centre-level and patient-level factors. 

Thank you for the opportunity to revise this manuscript. We thank the reviewers for their thoughtful and detailed comments, which we have addressed below.

Reviewer 1:

Author brought up the important concept to identify patient specific factors to target delivery of care. However, clarification is needed for their data and interpretation of their findings.

1) Effect of centre-level factors on HbA1c variation: Authors have identified that one reason might be that patient with stable type 2 diabetes are managed in primary care and more difficult cases are being referred to more specialized secondary or tertiary care centre. Authors have also cautioned about their finding in people with type 1 diabetes that patient volume association with her hemoglobin A1C was because tertiary care centres are located in more densely populated areas, where they see clinically complex patients.

There is no clear description of what is the available service at present to support authors’ interpretation. There is no analysis or data on what kind of care is delivered in these centres according to Chronic Care Model such as self-management support, clinical information system, or decision support. Another important information that will strengthen the paper is the frequency of contact with the health care providers, as well as ratio of patients and providers rather than volume of patients only. This information might also give the answer why diabetes centre location was not associated with variation in HbA1c and authors finding is different than other study. (Page 16-Line249)

Response: We thank the reviewer for this enquiry. A previous study has investigated differences in glycaemic control for people with type 2 diabetes using ANDA data and has found higher HbA1c in tertiary care centres compared to primary care. We have cited this paper (Lines 62-63 & 289-291, manuscript with track changes). 

With regard to the model of care, the ANDA diabetes centres were classified by their category within the NADC category model. These NADC categories of centre type represent the wide variety of diabetes centres across Australia, where people living with diabetes are treated in a range of primary, secondary and tertiary care settings. This paper presents analysis of an existing data set – a national clinical audit (ANDA). While providing an overview of the structure and the range of staff typically employed at each centre, discrete data about the numbers of staff, staff to patient ratio and model of care at each site is not available to us in the ANDA dataset. We suggest that future research may be helpful in elucidating the association of these factors with clinical outcomes, have noted this as limitation and have revised the manuscript accordingly (Lines 465-473 manuscript with track changes).

2)

Effect of patient-level factors on HbA1c variation

a) Hyperglycaemic emergencies and HbA1c variation: This paragraph is like a textbook. This needs revision to make relevant and explain authors finding.

Response: We appreciate this point and have revised the manuscript accordingly (Lines 305-327, manuscript with track changes)

b) The presence of diabetes complications and HbA1c variation: “The presence of diabetes complications was associated with higher HbA1c levels irrespective of diabetes type.” The logical explanation is higher HbA1C is associated with diabetes complication. It is important to identify what is the cause and what is the effect to appropriately plan care delivery.

Response: We agree with the reviewer that this is likely the case, but we are unable to infer causality from cross-sectional data as the temporal precedence of the risk factor in relation to outcomes cannot be determined, and this is not the purpose of this paper. We have highlighted this as a limitation of the study (Lines 472-473, manuscript with track changes)

c) Smoking status and HbA1c variation: Authors found that being a non-smoker was associated with reductions in HbA1c. People without diabetes in ANDA cohort also showed similar result. Authors finding sure emphasize importance of smoking cessation for everyone.

Response: With respect, as per line 197, all data in the ANDA sample is from people with diabetes. We therefore have not amended the manuscript in regard to this comment.

d) Aboriginal and Torres Strait Islander status and HbA1c variation; Since there is higher prevalence of type 2 diabetes in Aboriginal and Torres Strait Islander status, authors very appropriately suggested implementation of codeveloped resources and culturally and linguistically appropriate diabetes education programs. It will be important to know what is the current state of therapeutic relationship between the centers and these groups of patients. Is there any purposeful process of learning offered to the health-care workers to integrate specific contexts who are involved in care of these patients?

Response: We thank the reviewer for this thoughtful insight. Unfortunately, the ANDA data collection does not include details about the therapeutic relationship between providers and patients (ANDA data collection form – S 1). As such, we are unable to comment on whether learning is provided to healthcare providers who engage with Aboriginal or Torres Strait Islander communities. We agree that this is an important area, but it is outside the scope of this paper. We have highlighted the need for this work and suggested it as future direction (Lines 418-426, manuscript with track changes).

e) Glucose-lowering medications and HbA1c variation: It is not clear if insulin is included in the medications. Authors interpretation is “this is reflective of clinical practice where despite intensification of treatment and multiple pharmacological agents, some patients do not see clinically significant declines in HbA1c” (Page 20-line 330). Really? Could it be due to well-known providers’ “therapeutic inertia” - failure to initiate or intensify therapy in a timely manner resulting uncontrolled hyperglycaemia in patients with type 2 diabetes. At times multiple agents are added just to defer insulin initiation.

Insulin is included in the glucose-lowering medications, as per the ANDA data collection form (S 2). We have clarified this and have also taken the opportunity to address the potential issue of clinical inertia in this population (Lines 441-448, manuscript with track changes).

Reviewer 2

1. The authors have chosen to focus on HbA1C as the sole metric for quality of care and health improvement of those with diabetes. It was not (entirely) clear why the authors chose only HbA1C and this should better reconciled for the readership. Preferably please include (some pilot level) comparisons with fasting glucose, and/or post-prandial/diurnal glucose concentrations to demonstrate that HbA1C is the most sensitive marker in this cohort for the objectives. Are there validation data from this cohort elsewhere that the authors could utilize to strengthen this approach?

Response: We thank the reviewer for this suggestion. Unfortunately, as an annual cross-sectional benchmarking activity, we do not have the suggested auxiliary data for these patients. We have revised the manuscript to clarify the nature of ANDA and choice of HbA1c as an outcome of interest in this population (Lines 54, 62 – 65, manuscript with track changes).

2. For the international readership, please provide some scientific basis for why Aboriginal and Torres Strait Islander Peoples would be relevant to identify in the cohort.

Response: We have amended the manuscript to draw attention to the disparities in diabetes outcomes for Aboriginal and Torres Strait Islander peoples compared to the general population of Australia (lines 405-410, manuscript with track changes).

3. For clarification, please provide a more accurate description of the ‘Presence of diabetes complications’ as well as the extent of the ‘Hyperglycemic emergency episode(s)’, and include statements in the discussion and/or expand the interpretation based on this added information.

Response: We provide a description of the variables in the methods (Lines 142-160 manuscript with track changes), and have revised the discussion to more fully describe the ‘Presence of diabetes complications’ (Lines 330-333, manuscript with track changes) as well as suggested further work in this area (Lines 363-365, manuscript with track changes).

We have also revised the section in the discussion related to hyperglycaemic emergencies (Lines 305-326, manuscript with track changes).

4. It is known that increased adoption of behavioral change techniques by individuals increase the likelihood of smoking cessation. It is also plausible that individuals that are motivated to know/learn about health improvement will also be more motivated to implement self -care strategies to improve diabetes status and manage both glucose and HbA1C. Please consider these points in the discussion.

Response: We have revised the manuscript to reflect consideration of these points (Lines 368-387, manuscript with track changes). 

5. Please consider moving the ethical statement earlier(up) in the methods section.

Response: We have moved the ethical statement to Lines 102 – 112 (manuscript with track changes).

---

## [Editor Report · Decision Letter 1]

21 Jan 2022

Exploring HbA1c variation between Australian diabetes centres: the impact of centre-level and patient-level factors

PONE-D-21-14548R1

Dear Dr. Zoungas,

We’re pleased to inform you that your manuscript has been judged scientifically suitable for publication and will be formally accepted for publication once it meets all outstanding technical requirements.

Kind regards,

Spencer D. Proctor, PhD

Academic Editor

PLOS ONE
---

## [Editor Report · Acceptance letter]

27 Jan 2022

PONE-D-21-14548R1 

Exploring HbA1c variation between Australian diabetes centres: the impact of centre-level and patient-level factors. 

Dear Dr. Zoungas:

I'm pleased to inform you that your manuscript has been deemed suitable for publication in PLOS ONE. Congratulations! Your manuscript is now with our production department. 

Kind regards, 

on behalf of

Dr. Spencer D. Proctor 

Academic Editor

PLOS ONE